# Passive Phase Locking Coherent Combination of Solid-State Lasers through Stimulated Brillouin Scattering Effect

Yu Yu [1,2,*], Kai Li [1,2], Changyu Song [1,2], Hengzhe Yu [1,2], Di Wu [1,2], Zhipeng Xu [1,2], Yulei Wang [1,2] and Zhiwei Lu [1,2]

1    Center for Advanced Laser Technology, Hebei University of Technology, Tianjin 300401, China; 201911901011@stu.hebut.edu.cn (K.L.); 202021901040@stu.hebut.edu.cn (C.S.); 202311901009@stu.hebut.edu.cn (H.Y.); 202131903046@stu.hebut.edu.cn (D.W.); 202131903067@stu.hebut.edu.cn (Z.X.); wyl@hebut.edu.cn (Y.W.); zhiweilv@hebut.edu.cn (Z.L.)

2    Hebei Key Laboratory of Advanced Laser Technology and Equipment, Tianjin 300401, China

*    Correspondence: yuyu1990@hebut.edu.cn

**Abstract:** The stimulated Brillouin scattering (SBS) effect, a new approach to the combination of solid-state lasers, can be actualized via coherent synthesis. In this paper, a solid-state laser based on SBS passive phase locking, utilizing the master oscillator power amplifier (MOPA) structure at the front end of the lasers, provides the amplification of the Stokes light subsequently generated. In order to reduce the influence of thermal effects on beam quality, beam-split amplification has been adopted with the same phase locking used by the back injection of the Stokes pulse. With the advantage of the combined scheme, the energy extraction efficiency of SBS coherent combination can be reached at 91.8% with coherent fringe visibility of 83%. Therefore, it provides a new way to improve the brightness through realizing the coherent combination of multi-channel solid-state lasers.

**Keywords:** solid-state laser; stimulated Brillouin scattering; coherent combination





## 1. Introduction

High-energy pulsed laser systems are widely applied in laser processing, laser guidance, neutrino and proton generators, and laser fusion drives [1–4]. The application of chirp-pulse amplification technology has greatly promoted the development of short-pulse width and high-intensity laser technology, and the resulting extreme state conditions provide development opportunities for investigating the interaction between light and matter. However, due to the crystal growth process, the size of the laser crystal cannot be indefinitely increased, which limits the enhancement of laser output energy [5]. Furthermore, with the increase in the volume of the laser medium, the absorption efficiency of the pump has been refined. Unfortunately, the output power will inevitably produce a large amount of waste heat, which limits the improvement in beam quality and repetition frequency. For example, the master oscillation power amplifier (MOPA) is used to amplify a beam of seed signal light with high beam quality through each MOPA unit step by step. And the amplified energy and beam quality are limited by the working state of each stage amplification unit, that is, the failure of any stage amplification unit will affect the laser output and even damage the laser system. Therefore, it is difficult to achieve a laser system that meets the same high power, high beam quality, and high reliability laser output by solely relying on laser oscillator or power amplifier technology. Thermal lensing, thermo-optical wedge, and thermally induced birefringence effects are inevitable, which lead to a serious decrease in the average power of the output laser and the deterioration of the beam quality [6,7].

The coherent synthesis of multiple lasers is one of the most effective ways to obtain high-brightness laser output, and it has gradually become a research hotspot in the field of laser technology [8–10]. In order to achieve higher brightness output at high repetition rates, high-energy laser systems must maintain good beam quality while increasing the laser

output power. At present, the device mainly uses multi-channel laser synthesis focusing to improve the output energy, especially when using coherent synthetic focusing; it can double the power density of the focal spot to provide ultra-high power density electric field extreme state conditions required for physical research, which will be an important research direction in the field of high-energy short-pulse lasers in the future. However, how to solve the phase locking, co-frequency resonance, and thermal management problems between multiple beams of light with independent optical parameters (frequency, vibration direction, polarization state and phase) in coherent synthesis laser systems are two key technical problems involved in solid-state laser coherent synthesis systems [8].

The coherent beam needs to meet the four conditions of consistent center frequency, narrow linewidth, same polarization direction, and good beam quality of the laser source. Moreover, the directivity of each laser is required, and, finally, the phase lock of each beam is required [9]. The above points, as long as one of them is not performed well, will affect the effect of coherent synthesis. Depending on the phase-locking method, coherent synthesis techniques can be divided into two categories: passive coherent synthesis [10] and active coherent synthesis [11]. Passive coherent synthesis realizes the phase locking of multiple lasers inside of the laser through the principle of self-organization, and its implementation methods mainly include the interferometer mode [12], non-linear self-organization mode [13], and evanescent wave coupling method [14]. Active coherent synthesis uses a certain control algorithm to realize the phase locking of multiple lasers outside of the laser through the phase modulation device, and the control methods used mainly include the heterodyne method [15], adaptive phase-locked method, etc. [16]. Based on corner cube, Cheng Yong. et al. [17] utilized mutual injection phase-locked technology to realize six-channel laser coherent synthesis. It exhibited that the energy of the output light is 15.3 J with a pulse width of 500 µs, the divergence angle is 1.7 mrad, and the synthesis efficiency is as high as 95.6%. Utilized self-phase-locking technology, Hongjin Kong et al. [18] adopted the wavefront segmentation of pump pulses and combining with SBS-PCM. Using piezoelectric ceramic (PZT) feedback to adjust the distance between the concave mirror and the SBS cell, the coherent synthesis of four beams was achieved. However, it still requires the detection and modulation of the phase of the laser array and the real-time adjustment of the PZT feedback mirror at the end of the SBS-PCM in each laser path based on the test and calculation results to achieve coherent laser output, which greatly increases the complexity of the locking system.

Compared to the active phase-lock technology, more concisely, the passive method does not need a complex circuit phase control system, including a phase detection device and strict and complex algorithms. The overall structure is complex and needs to occupy more space, and passive phase-locking technology is a method of self-phase adjustment. Through the structure of some optical structures or optical characteristics, finally, it realizes the phase-locking method of two beams or multiple beams. Therefore, we demonstrate a passive phase-locking coherent combination method of solid-state lasers through a stimulated Brillouin scattering effect. Meanwhile, it paves the way for the excellent performances of lasers, including high pulse energy and repetition frequency.

## 2. Materials and Methods

The phase difference can be determined based on whether the two Stokes beams have been coherent or not and the passive phase modulation method has been adopted. Simply, it only needs a passive adaptive system and detailed settings for the overall optical path. The passive phase-locking method used in this experiment was used to output Stokes light using stimulated Brillouin scattering (SBS). Though back-injecting seed light, the phases of the two Stokes beams could be locked with the same phase of the seed beam. In the meanwhile, the coherent synthesis of two Stokes beams was to control the phase difference. And the phase difference could be measured via the coherent combination fringes by adjusting the optical path. And the phase difference could be adjusted based on the distance of the optical path. For more detail, the closer the optical path was to the phase

difference of the two beams, the smaller the phase difference. Coherent fringe visibility, as the vital parameter used to characterize coherent combination, was produced by the two Stokes beams after an encounter.

According to the coupled wave equation [19], the electric field components of pump and Stokes light can be expressed via the following equations.

$$\frac{\partial E_L}{\partial Z} + \frac{n}{c}\frac{\partial E_L}{\partial t} = \frac{i\omega_L\gamma}{2nc\rho_0}\rho E_S - \frac{\alpha}{2}E_L \tag{1}$$

$$-\frac{\partial E_S}{\partial Z} + \frac{n}{c}\frac{\partial E_S}{\partial t} = \frac{i\omega_S\gamma}{2nc\rho_0}\rho^* E_L - \frac{\alpha}{2}E_S \tag{2}$$

$$\frac{\partial^2\rho}{\partial t^2} - (2i\omega - \Gamma_B)\frac{\partial\rho}{\partial t} - (i\omega\Gamma_B)\rho = \frac{\gamma}{4\pi}q_B^2 E_L E_S^* \tag{3}$$

where $n$ is the refractive index of SBS medium, and $E_L$ and $E_S$ represent the amplitudes of pump and Stokes. And $\omega_L$ and $\omega_S$ are the corresponding angular frequencies. $\rho$, $q_B$, c, $\gamma$, $\rho_0$, $\Gamma_B$, and $\alpha$ are the density amplitude in SBS medium, the wave vector of the acoustic field, the speed of light, electrostrictive constant, and the density under medium equilibrium state, respectively.

$$E_{Lj+1}^{m+1} = [K(p_{1j}^m - p_{2j}^m) * E_{Sj}^{m+1} + (R-1)E_{Lj}^{m+1} - \frac{\Delta t}{2}\alpha E_{Sj}^{m+1} + E_{Lj}^m]/R \tag{4}$$

$$E_{Sj}^{m+1} = [-K(p_{1j}^m - p_{2j}^m) * E_{Sj}^{m+1} + RE_{Lj}^{m+1} + E_{Sj}^m]/(1 + R + \frac{\Delta t}{2}\alpha) \tag{5}$$

In order to achieve better Stokes light waveform output, the generation cell was simulated. In particular, the vital parameters for FC-770 include the medium refractive index of 1.27, phonon lifetime of 0.57 ns, gain coefficient of 3.5 cm/GW, and the Brillouin frequency shift of 1081 MHz, respectively. And the simulation results are shown in Figure 1, and the waveform of the output Stokes light was obtained by changing the energy injected into the media cell due to the returned Stokes pulse. The light was amplified by extracting the injected pump light with the same, meaning that the output Stokes light produced a steep front, and as the energy was extracted, the energy of the pump light decreased, meaning that the waveform of the output Stokes light produced a certain degree of tailing phenomenon.

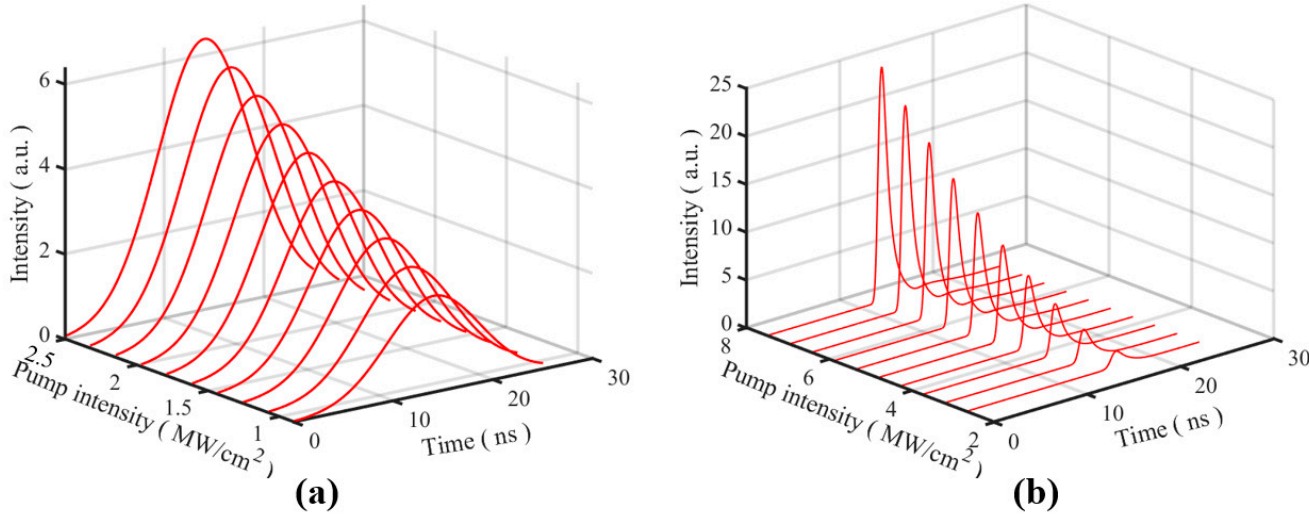

**Figure 1.** Simulated input light and output optical waveform: (**a**) input pump optical waveform and (**b**) output Stokes optical waveform.

As shown in Figure 2, with the increase in the light energy injected into the pump, the returned Stokes light energy became stronger and stronger, and its energy reflectivity gradually increased. With the further improvement in energy injection, the trend of energy extraction efficiency will further increase.

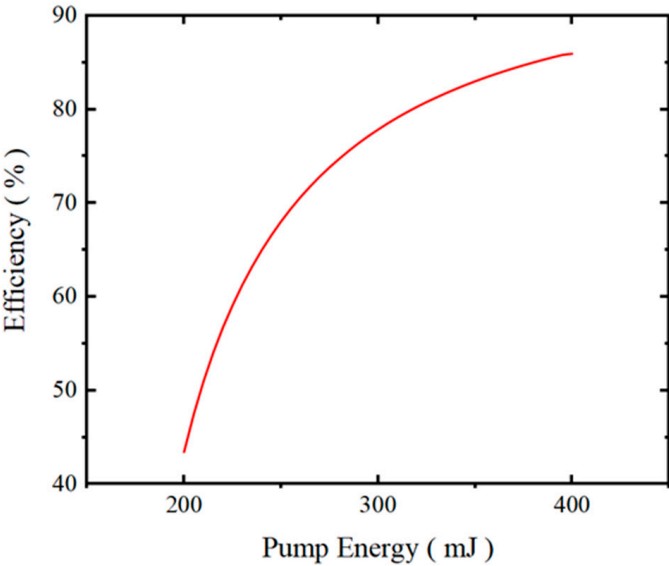

**Figure 2.** Stokes light reflectance curve with the relationship of pump energy.

In the simulation, the waist radius of a single laser was 9 mm, the output wavelength of the sub beam was 1064 nm, the input power of each sub beam was 10 W, and the operating mode was $TM_{00}$. In order to move closer to the experiment, observation distances were set at 100 m, 200 m, and 300 m, respectively. The interval between the two sub beams was 2 to 14 mm. The interval distance was 4 mm, and the spot pattern, main peak width, peak light intensity, etc. after beam merging were recorded. As shown in Figure 3, if the distance between the two pump lasers was close enough, the two pump beams were completely merged into one beam output, and the light intensity of the output laser was the strongest point. In the meanwhile, if the distance between the two beams of light was close enough, the energy of the two pump beams could be completely synthesized into one combined beam. It indicated that the distance between the two beams of pump lights had a more direct impact on the main peak output light intensity. With the increase in the interval between the two optical beams, interference fringes gradually appeared. Therefore, the further the distance between the two beams, the phenomenon with more interference fringes increased, and the peak power of the main peak output decreased.

The structure of the main oscillation power amplifier was used, as shown in Figure 4. And the whole device consisted of a seed source, a phase conjugate mirror dielectric cell, a unidirectional single-channel amplifier, a stimulated Brillouin collinear amplification dielectric cell, a beam synthesis system, and multiple isolators. The seeder could output an energy of 600 mJ, a pulse width of 10 ns, and a wavelength of 1064 nm. The resonant reflector used in the three-side resonance was composed of two mirrors, with different transmittances as the output mirror. This structure could have a strong inhibitory effect on the neighboring resonant cavity due to the use of a two-part transmittance coupling mirror. On the other hand, it increased the output energy of the resonant cavity, allowing the laser to withstand greater energy. The working principle of a torsion pendulum cavity was to convert the linearly polarized light and circularly polarized light in the cavity into each other through two quarter-wave plates. It allowed the laser beam to propagate uniformly in the gain medium, thereby suppressing the spatial hole burning effect. For the laser beam, the amplification method mainly involved using a single lamp pump amplification method. In the state of lamp pumping, the cooling effect on both sides of the laser medium was

uneven, meaning that the crystal produced a thermal effect, which would have a great impact on the beam quality. Therefore, in order to reduce the impact of the amplified thermal effect on the beam quality, the seed source amplifier and the two single lamp pump amplifiers were placed in opposite directions. And the laser underwent deformation after the first stage of amplification. The amplification was compensated after the second stage of amplification, meaning that the output of the amplified laser energy was distributed as uniformly as possible. The laser produced by the seed derived laser was divided into two beams through a splitter mirror. The first beam was used as pump light to produce Stokes light, and the second beam was amplified at an amplification stage to increase its energy and used to amplify Stokes light.

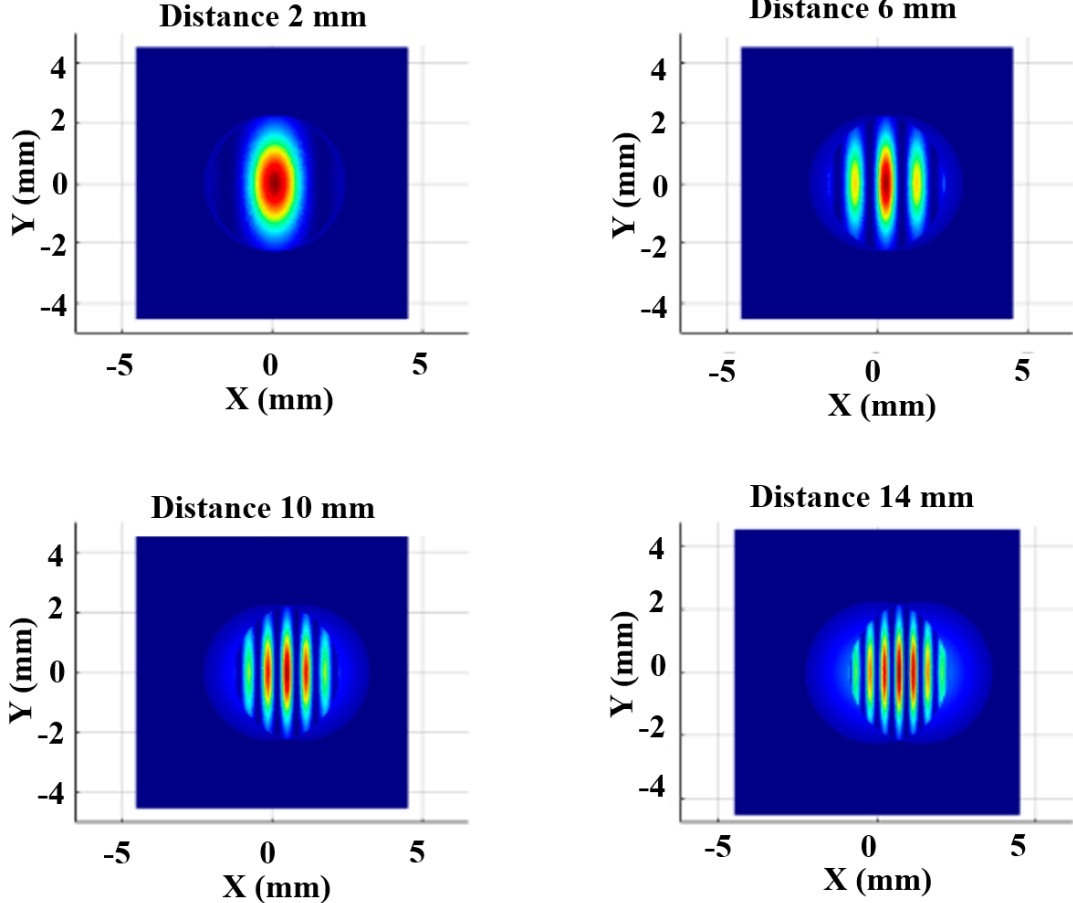

**Figure 3.** Coherent synthesis of two beams with different distances.

In the experiment generating the Stokes beam, FC-770 was used as the generation medium [5], the structure of generating the Stokes beam was a single SBS cell structure, and the window mirror of the traditional media cell was adopted with the same parameters as the simulation part. Although this structure could ensure that most of the laser energy passes through the window mirror and would not produce large refraction after passing through the window mirror, due to the limitation of the coating process, the pump light energy could not completely pass through the medium cell, and some of the light was still be partially reflected by the window mirror. The surface was reflected, and although the reflected pump light energy was low, it still affected the returned Stokes beam, resulting in an inability to accurately measure the energy and beam profile of the Stokes light. Therefore, a new type of media cell structure was adopted, that is, the glass tube with a certain inclination angle on the end face was used as the Stokes light production pool. After that, the pump light reflected by the window surface would be reflected to the optical platform and would not be returned to the incident light path by the window mirror. The

transmitted pump light would produce Stokes light through the action of SBS, and Stokes light would be reflected and output along the incident light path, and in this way, it would effectively work. In addition, the effect of the returned pump light on the returned Stokes light was noted.

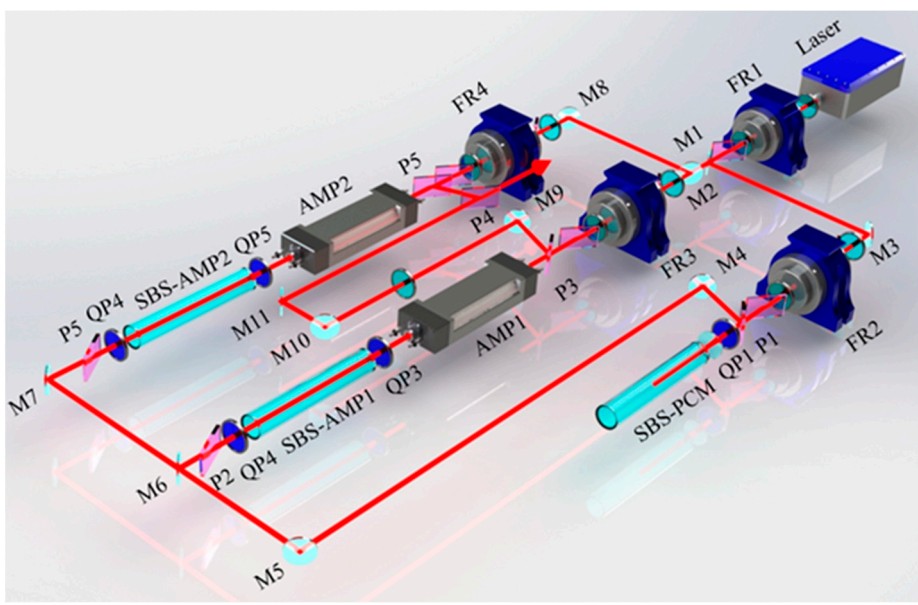

**Figure 4.** Scheme of experimental setup (the abbreviated components depict as follows, FR: Faraday rotator; AMP: amplifier; QP: quarter-wave plate; P: polarizer).

Therefore, the seed light path was adjusted as follows: the seed passed through the beam splitter mirror. In particular, the first beam of light passed through the isolation system, and after passing through a quarter-wave plate, the P polarized light was converted into circularly polarized light and focused into the SBS cell through a lens, and SBS occurred in the cell to produce a circularly polarized Stokes light with high beam quality. After passing through a half-wave plate again, the Stokes light was converted into P-polarized light, and it was again divided into two beams by a beam-splitter (T/R = 5:5). After passing through a quarter-wave plate, P polarized light was converted into circularly polarized light. Injected into two SBS media cells, it absorbed the amplified pump energy and passed through a quarter-wave plate again to convert circularly polarized light into the S polarized state again. The polarized light was output from the entire optical path system, and the output light was adjusted through the optical path and injected into the same polarizer at the same time for the final polarization beam output.

Originally, the Stokes beam from the seed source was divided into two beams by a beam-splitter (T/R = 1/1), and two beams of light were amplified through different amplifiers. After passing through a polarizer and a quarter-wave plate, the light changed from P polarization state to S-polarized light. It was injected into the SBS medium pool to generate Stokes light. The pump was amplified by different amplifiers, and the amplified pump was converted into circularly polarized light by a quarter-wave plate in the media cell to achieve collinear amplification. And the pump light that could not be extracted passed through the SBS media cell. After being converted into S-polarized light through a quarter-wave plate again, it was reflected out of the optical path by the polarizer and entered the light absorption system. The amplified Stokes beam was converted into S-polarized light by a quarter-wave plate, reflected through the polarizer, and output through the final beam clamping system.

## 3. Results and Discussion

The beam combination system in the experiment consists of a mirror, a half-wave plate and a polarizer. In detail, the half-wave plate is used to adjust the polarization state of each beam. Functionally, it can convert the S-polarized light into P-polarized light. After being injected into the polarizer, together with the unconverted S polarized light, the combined beam output with the same phase and the two Stokes beams after amplification are injected in the polarizer. The optical path difference can be adjusted by a set of coupling mirrors, which are fixed onto a set of sliding tables. In order to manipulate the optical path difference, the coupling lens group is adjusted by moving the position of the frame on the slide table. Therefore, to change the optical path difference and adjust the two beams of light, the optimal beam has been fixed through the position of the coupling lenses.

Figure 5a shows the change in Stokes light energy reflected back with the increase in the pump light energy injected into the generation SBS cell. In order to achieve high energy reflection, the focal length of the lens and the length of SBS cell are fixed at 80 cm and 60 cm, respectively. When the pulse width of the injected pump light is 10 ns, the reflectivity of the injected pump light increases with the increase in injected energy. With 400 V of input voltage, the pump light energy is 51 mJ, the reflected Stokes light energy is 40.3 mJ, and the energy reflectivity is about 77.1%. When the input voltage is set to 900 V, the pump energy is 428 mJ, the Stokes energy is 388 mJ, and the energy reflectivity is 90.7%. Moreover, the beam profile of the injected pump light was improved to some extent via the action of SBS-PCM, and the energy distribution of the laser spot output became more uniform [20,21].

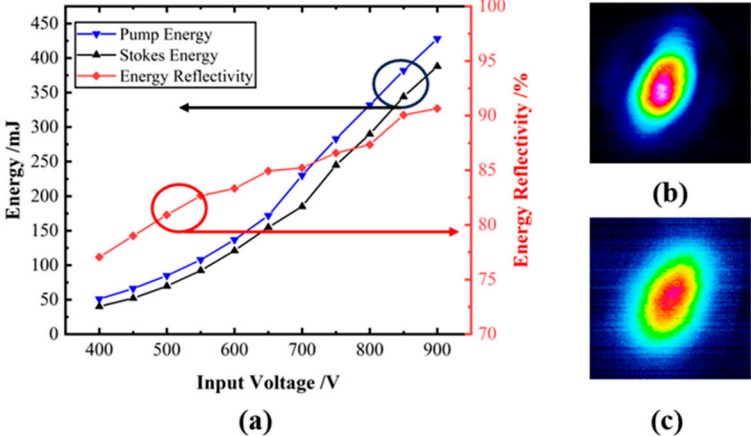

**Figure 5.** (**a**) The Stokes light energy and energy reflectivity after the generation of the SBS-PCM phase conjugate mirror, (**b**) injected pump spot, and (**c**) output Stokes spot.

In the Stokes light amplification experiment, the generated Stokes beam is separately injected into two amplification cells for amplification, and the injected Stokes light is amplified by extracting the energy from the injected pump light; the energy change in the two is shown in Figure 6. The Stokes beams are amplified by extracting the energy of the pumped light. The energy changes in the two are shown in Figure 6. Figure 6a,b show the magnification curves of the first and second beams, respectively. Due to the influence of thermal effect, the laser center of the amplified pump becomes unstable. As a result, Stokes light cannot extract more pump energy, and the extraction efficiency of Stokes light energy becomes lower and lower. Finally, the energy extraction efficiency of the two channels decreased from the maximum of 85.3% and 85.9% to 71.8% and 71.7%, respectively, and the final output energy increased from 446 mJ and 474 mJ to 800 mJ and 797 mJ, respectively.

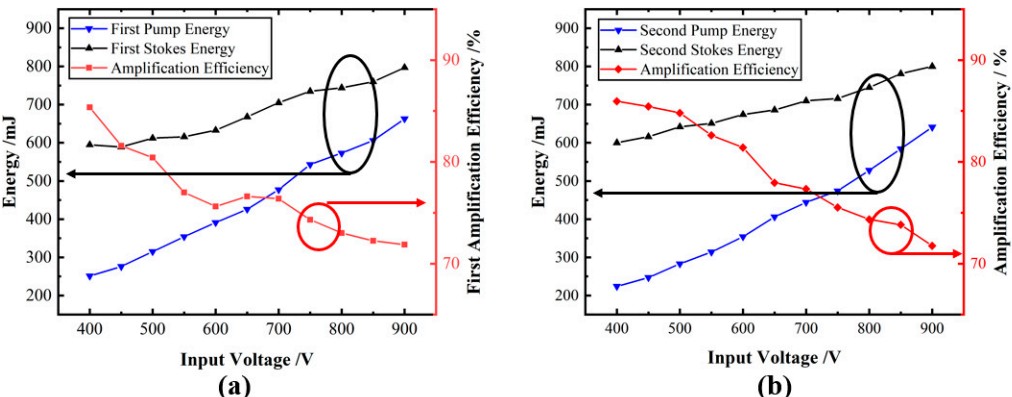

**Figure 6.** The first (**a**) and second (**b**) laser energy and energy extraction efficiency with the relation of input voltage.

The coherent beam combining with SBS passive phase-locked technology, as one of the self-organizing phase-locked and overlapping coupling beam-combining methods, plays a vital role in achieving mutual influence between multiple lasers and phase-matching conditions. However, an increase in the number of lasers will inevitably lead to an increase in the energy injected into the medium pool, and an increase in energy density will increase the probability of electrical breakdown. It causes the quality of the combined output laser beam to deteriorate. The backward seed injection passive phase-locked beam-combining structure ensures that the phase of the signal light does not change during amplification, playing a stable phase difference role. Moreover, the use of the backward seed injection structure can effectively reduce the breakdown threshold, thereby achieving higher energy extraction efficiency. The use of phase conjugation mirror technology can effectively control the phase difference of the generated signal light. By organizing all of the above methods, this experiment proposes to use phase conjugation mirror technology to compensate for the wavefront distortion generated by the pump light and use the characteristics of phase conjugation mirror to achieve the phase locking of the signal light. Using the wavefront-splitting method, the signal light is divided into two beams, which are then injected into the SBS medium pool as seed light to achieve re-amplification. Finally, the beam-combining device is used to achieve beam-combining output.

As shown in Figure 7, to further evaluate the beam quality of coherent synthesis, two beams after coherent combination with the condition of SBS phase locking in the near beam (a) and the far beam (b) and the interference fringes (c) without phase locking have been measured. Indeed, through the backward injection of Stokes beams, the phase difference between two beams with the approximately equal beam optical path has been compensated for, satisfying the phase-locking condition. In this way, the phase difference between the two beams can be further controlled; therefore, two beams can be maintained through phase locking through backward injection. Moreover, the laser output pulse energy of 1245 mJ with the repetition frequency of 10 Hz has been obtained through coherent combination. Meanwhile, the output energy of the two beams behind the polarizer is 652 mJ and 704 mJ, respectively, and the beam combining efficiency is as high as 91.8%. The reason for this outcome is that SBS effect can not only satisfy phase locking above the injected seed light phase, but can also compensate for thermal distortion. In particular, with the assistance of backward injection, coherent synthetic lasers exhibit high stabilities, including pulse energy, beam quality, and directional stability. As the vital parameter to character coherent combination, the coherent fringe visibility of the two beams has been measured as high as 83%.

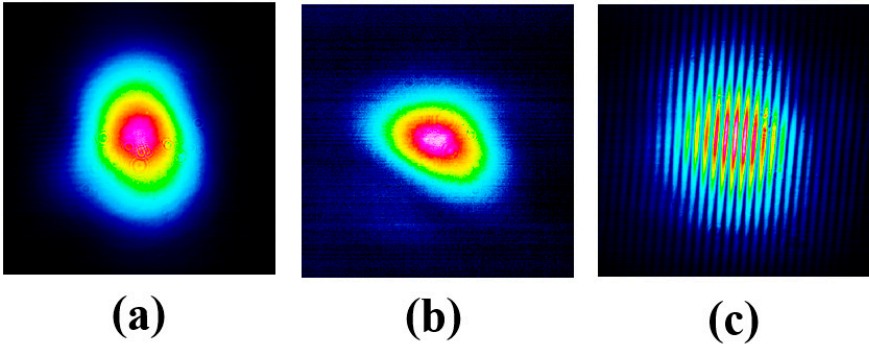

**Figure 7.** (**a**) Synthetic beam near-field spot (**b**) Synthetic beam far-field spot (**c**) Synthetic light interference fringe without phase-locked coupling.

## 4. Conclusions

In this paper, a solid-state laser based on stimulated Brillouin phase-locked coupling was developed, the amplification mode adopted the collinear amplification mode was used, and the pump participating in beam binding was matched via SBS-PCM phase conjugate mirror technology. The light is optimized to make the energy distribution inside of its spot more uniform, and the resulting high-beam quality Stokes light is divided into two beams of similar energy through the beam-splitter to participate in amplification, passing through the SBS cell. The collinear amplification method achieved more than 70% energy extraction efficiency, and 1.245 J was obtained via the coherent combination. With the increase in the beam quality of the laser output, the coherent fringe visibility of the two beams of light can reach 83%.

**Author Contributions:** Conceptualization, Y.Y. and K.L.; methodology, C.S.; formal analysis, Z.X.; investigation, H.Y.; resources, Z.X.; data curation, H.Y. and D.W.; writing—original draft preparation, Y.Y. and C.S.; writing—review and editing, H.Y.; visualization, Z.L.; supervision, Y.W.; funding acquisition, Y.Y. All authors have read and agreed to the published version of the manuscript.

**Funding:** This work was supported by the National Natural Science Foundation of China (No. 62005074, No. 62004059, No. 61927815, and No. 62075056), the Natural Science Foundation of Hebei Province (No. F2021202002, and No. F2022202035), and the Foundation of the Anhui Laboratory of Advanced Laser Technology (AHL2021KF01).

**Institutional Review Board Statement:** Not applicable.

**Informed Consent Statement:** Not applicable.

**Data Availability Statement:** Not applicable.

**Conflicts of Interest:** The authors declare no conflict of interest.

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
