# Peer review of "Passive Phase Locking Coherent Combination of Solid-State Lasers through Stimulated Brillouin Scattering Effect"

_photonics, doi:10.3390/photonics10101098_

Round 1
Reviewer 1 Report
The paper reports research in coherent combination of a joule-level pulsed nanosecond laser that has been conditioned in two liquid Brillouin amplifiers. The light is optimized to make the energy distribution inside its spot more uniform, and the resulting high beam quality Stokes light is divided into two beams of similar energy through the beam-splitter to participate in amplification, passing through the SBS. The collinear amplification method achieved more than 70% energy extraction efficiency, and 1.245 J was obtained by the coherent combination. With the higher beam quality of the laser output, the coherent fringe visibility of the two beams of light can reach 83%. It can be published after few minor revisions.
1. The sentence of “The beam combination system in the experiment consists of a mirror, a half waveplate and a polarizer, and the half-wave plate is used to adjust the polarization state of one of the beams, convert the S-polarized light into P-polarized light, inject it into the P-larizer together with the unconverted S polarized light, and combine the beam output through the polarizer, and the two beams of light are injected into the position sum in the polarizer.” is too long. It Should be rewritten into a few short sentences.
2. The “Pump light” with upper text and “pump light” with lower text in the whole manuscript should be same.
3. The SBS process requires the corresponding light source with narrow line width, the physical mechanism should be discussed in detail.
Author Response
Dear Reviewer,
Thank you very much for your letter and advice on our manuscript. It is important to improve the manuscript. Accordingly, we have revised the manuscript. All amendments are underlined in the revised manuscript. In addition, point-by-point responses to the comments are listed below this letter.
We hope that the revised version of the manuscript is now acceptable for publication in your journal. I’m looking forward to hearing from you soon.
Sincerely,
Yu Yu
We would like to express our sincere thanks to the reviewers for the constructive and positive comments.
Replies to the reviewers:
Comment 1: " The sentence of “The beam combination system in the experiment consists of a mirror, a half waveplate and a polarizer, and the half-wave plate is used to adjust the polarization state of one of the beams, convert the S-polarized light into P-polarized light, inject it into the P-larizer together with the unconverted S polarized light, and combine the beam output through the polarizer, and the two beams of light are injected into the position sum in the polarizer.” is too long. It Should be rewritten into a few short sentences. "
Reply: Thanks for your suggestions. We have rewritten into a few short sentences in the revised manuscript. In detail, “The beam combination system consists of a mirror, a half waveplate and a polarizer. The half-wave plate is used to adjust the polarization state of the two beams with the function of injecting it into the P-larizer together with the unconverted S polarized light. In the meanwhile, it can combine the beam output through the polarizer, and the two beams of light are injected into the position sum in the polarizer.”
Comment 2: “The “Pump light” with upper text and “pump light” with lower text in the whole manuscript should be same.”
Reply: Thanks for your suggestions. We have revised the “Pump” into “ pump” in the whole revised manuscript.
Comment 3: “The SBS process requires the corresponding light source with narrow line width, the physical mechanism should be discussed in detail.”
Reply: Thanks for your advice. The SBS process should be occur with narrow line width pump light, as the reviewer’s points. By using the theory of plane mirror mode selection with three plane resonance (transmittance of 90%) in a flat cavity, the laser can output with single longitudinal mode.

Reviewer 2 Report
The authors developed a solid-state laser based on stimulated Brillouin phase-locked coupling. In order to reduce the influence of thermal effects on beam quality, beam split-amplification has been adopted with phase locking by the back injection of Stokes pulse. Utilizing the advantage of combined scheme, the energy extraction efficiency of SBS coherent combination is stated to be >90%. The paper may be interesting to the specialists in the field. But I have the following comments to the authors.
The main concerns are as follows.
1. It is not clear how curves in Figures 1 and 2 are calculated. The numerical model must be briefly discussed and master equations must be added.
2. It is seen that experimental beam profiles are not ideal, but strongly skewed. What is about M-factors? I have also some doubts relative to the statement “beam combining efficiency is high to 91.8%”. More convincing evidence should be provided.
The minor points.
1. I am not a native speaker, but many phrases seem grammatically incorrect and very difficult to understand.
2. The paper does not fully match to the template.
3. Figure 3 has very poor quality.
4. Line 30, “main oscillation power amplifier (MOPA)”. The commonly used meaning of MOPA is master oscillation power amplifier.
5. There are a lot of inaccuracies, for example:
Line 48, “…laser in the future 。 However,…”
Line 91, “returned Stokes The light is amplified”
Line 108, “light intensity With the increase”
In many cases, “Pump” should be replaced by “pump”.
Author Response
Dear Reviewer,
Thank you very much for your letter and advice on our manuscript. It is important to improve the manuscript. Accordingly, we have revised the manuscript. All amendments are underlined in the revised manuscript. In addition, point-by-point responses to the comments are listed below this letter.
We hope that the revised version of the manuscript is now acceptable for publication in your journal. I’m looking forward to hearing from you soon.
Sincerely,
Yu Yu
We would like to express our sincere thanks to the reviewers for the constructive and positive comments.
“The authors developed a solid-state laser based on stimulated Brillouin phase-locked coupling. In order to reduce the influence of thermal effects on beam quality, beam split-amplification has been adopted with phase locking by the back injection of Stokes pulse. Utilizing the advantage of combined scheme, the energy extraction efficiency of SBS coherent combination is stated to be >90%. The paper may be interesting to the specialists in the field. But I have the following comments to the authors.”
Replies to the reviewers:
Comment 1: “It is not clear how curves in Figures 1 and 2 are calculated. The numerical model must be briefly discussed and master equations must be added.”
Reply:
Comment 2: “It is seen that experimental beam profiles are not ideal, but strongly skewed. What is about M-factors? I have also some doubts relative to the statement “beam combining efficiency is high to 91.8%”. More convincing evidence should be provided.”
Reply: Thanks for your suggestions. We have added more explanation for simulation part. According to the coupled wave equation [Damzen M, Vlad V, Mocofanescu A, and Babin V. Stimulated Brillouin scattering: fundamentals and applications[M]. CRC press, 2003], the electric field components of pump and stokes light can be expressed as the following equations.
(1)
(2)
(3)
where, n is the refractive index of SBS medium, EL and ES represent amplitudes of pump and stokes. And ωL and ωS are corresponding angular frequency angular frequencies. ρ, qB, c, γ, ρ0, ΓB, and α are the density amplitude in SBS medium, the wave vector of the acoustic field, the speed of light, Electrostrictive constant, the density under medium equilibrium state.
(4)
. (5)
The numerical calculation formula can be obtained through the coupled wave equation. In order to achieve better Stokes light waveform output, the generation cell was simulated, and the simulation results are shown in Figure 1.
Comment 3: “I am not a native speaker, but many phrases seem grammatically incorrect and very difficult to understand.”
Reply: Thanks for your suggestions. We have rewritten into right styles in the revised manuscript.
Comment 4: “The paper does not fully match to the template.”
Reply: Thanks for your suggestions. We revised the style of manuscript as the template of Photonics to match template.
Comment 5: “ Figure 3 has very poor quality”
Reply: Thanks for your suggestions. The original figure 3 has been replaced as the new one with high quality in the revised manuscript.
Comment 6: “Line 30, “main oscillation power amplifier (MOPA)”. The commonly used meaning of MOPA is master oscillation power amplifier.”
Reply: Thanks for your suggestions. We have revised it in the revised manuscript.
Comment 7: “There are a lot of inaccuracies, for example: Line 48, “…laser in the future 。 However,…” Line 91, “returned Stokes The light is amplified” Line 108, “light intensity With the increase” In many cases, “Pump” should be replaced by “pump”.”
Reply: Thanks for your suggestions. We have revised it in the revised manuscript. In detail, “laser in the future. However,” “the returned Stokes pulse. The light is a” “light intensity. With the increase” And “Pump” should be replaced by “pump” in the revised manuscript.

Reviewer 3 Report
In order to reduce the influence of thermal effects on beam quality, Yu et al. adopted a beam split-amplification approach with the same phase locking by back injecting a Stokes pulse. With the advantage of this combined scheme, the energy extraction efficiency of SBS coherent combination can reach 91.8% with a coherent fringe visibility of 83%. This paves a new way to improve brightness through the realization of coherent combination of multi-channel solid-state lasers. The paper can be published in Photonics after some minor revisions. 1) In the experiment to generate the Stokes beam, FC-770 was used as the SBS generation medium. The authors should provide a reason for this choice. 2) The authors report that if the pump energy is 428 mJ, the returned Stokes energy is 388 mJ, resulting in an energy reflectivity of 90.7%. This high energy reflectivity should be explained by the authors. 3) To reduce the influence of thermal effects on beam quality, beam split-amplification with the same phase locking by back injecting a Stokes pulse has been adopted. The authors should provide a clearer explanation of the passive phase-lock effect.No
Author Response
Dear Reviewer,
Thank you very much for your letter and advice on our manuscript. It is important to improve the manuscript. Accordingly, we have revised the manuscript. All amendments are underlined in the revised manuscript. In addition, point-by-point responses to the comments are listed below this letter.
We hope that the revised version of the manuscript is now acceptable for publication in your journal. I’m looking forward to hearing from you soon.
Sincerely,
Yu Yu
In order to reduce the influence of thermal effects on beam quality, Yu et al. adopted a beam split-amplification approach with the same phase locking by back injecting a Stokes pulse. With the advantage of this combined scheme, the energy extraction efficiency of SBS coherent combination can reach 91.8% with a coherent fringe visibility of 83%. This paves a new way to improve brightness through the realization of coherent combination of multi-channel solid-state lasers. The paper can be published in Photonics after some minor revisions.
Comment 1: “In the experiment to generate the Stokes beam, FC-770 was used as the SBS generation medium. The authors should provide a reason for this choice.”
Response: Thanks for your suggestions. In order to achieve better Stokes light waveform output, the generation cell was simulated. Particularly, the vital parameters for FC-770 include medium refractive index of 1.27, phonon lifetime of 0.57 ns, gain coefficient of 3.5 cm/GW and the Brillouin frequency shift of 1081 MHz, respectively. And the simulation results are shown in Figure 1, and the waveform of the output Stokes light is obtained by changing the energy injected into the media cell, due to the returned Stokes pulse. The light is amplified by extracting the injected pump light with the same, so the output Stokes light produces a steep front, and as the energy is extracted, the energy of the pump light de-creases, so that the waveform of the output Stokes light produces a certain degree of tailing phenomenon.
Comment 2: “The authors report that if the pump energy is 428 mJ, the returned Stokes energy is 388 mJ, resulting in an energy reflectivity of 90.7%. This high energy reflectivity should be explained by the authors.”
Response: Thanks for your suggestions. With the increase of the light energy injected into the pump, the returned Stokes light energy is getting stronger and stronger, and its energy reflectivity is gradually in-creasing. With the further improvement of energy injection, the trend of energy extraction efficiency is still increasing.
Comment 3: “To reduce the influence of thermal effects on beam quality, beam split-amplification with the same phase locking by back injecting a Stokes pulse has been adopted. The authors should provide a clearer explanation of the passive phase-lock effect”
Response: Thanks for your suggestions. Compared to the active phase-lock technology, more concisely, the passive method does not need a complex circuit phase control system including phase detection device, strict and complex algorithms. The overall structure is complex and needs to occupy more space, and passive phase-locking technology is a method of self-phase adjustment. Through the structure of some optical structures or optical characteristics, and finally it realizes the phase-locking method of two beams or multiple beams. Therefore, we demonstrate a passive phase locking coherent combination method of solid lasers through stimulated Brillouin scattering effect. Meanwhile, it paves the way for excellent performances of lasers including high pulse energy, and repetition frequency.

Reviewer 4 Report
Comments:
A major problem is in this paper is too long sentences. A lot of information is put into one sentence and it’s difficult to follow through. Break the sentences so that there is clarity and helps in understanding the paper. Also, the authors have written pump with capital P in the middle of sentences which is not required. There are places where phrases have been repeated (for example: lines 76-77). So, it would be better if they can re-write the paper with shorter sentences and include more details.
Lines 8,10: Stimulated Brillouin scattering instead of Brillouin scattering. MOPA is master oscillator power amplifier. Please correct it in the text. Also, I have noticed that the authors have repeated stimulated Brillouin scattering throughout the text. Once you have mentioned the abbreviated form (such as SBS or MOPA), you can simply continue writing the abbreviated form instead of the full form in the subsequent text.
In the introduction, the sentences are too long. Please break the sentences for more clarity.
Line 37: What is thermo-optical wedge?
Line 40: Please provide references.
Line 49: what is co-frequency resonance?
Line 65: Please write “compared to” instead of “compared with”.
Lines 68-71: The advantages for passive phase-locking needs more clarity. It would also be better if shorter sentences are used to describe.
Line 73: A reference for coherent synthesis would be good.
Lines 77-81: Sentence is too long and needs clarity. Also, the comparison of both the techniques is repeated and not required.
Line 88: The simulation part needs more explanation. What equations/parameters were used to simulate? How does the generation cell look like?
Fig. 2: How is the Stokes light reflectance calculated? What is the reason for the shape of the curve?
Fig.3: A diagram to explain the simulation would be helpful.
Lines 116-122: Too long a sentence. Please break it to multiple sentences.
Fig. 4: Mark the beam direction. Please mention what each abbreviated component depicts (such as FR: Faraday rotator…).
It is difficult to follow through the experimental setup text since the authors have not specified which mirrors, generator cells they are talking about. Please specify the components in the description so that the reader can understand. For example, in line 48: “Therefore, the seed light path is adjusted as follows, the seed passes through the beam splitter mirror (M1)”.
Also, the setup lacks details. The authors need to describe the length of the generator cells. FC 770 is used as the SBS medium. A reference is needed for the material. Please specify the Brillouin shift and expected Stokes wavelength. What are AMP1 and AMP2? How does SBS-PCM works? Please provide references for it too.
Lines 139-144: Was this method used to separate the pump and the Stokes beam? If that is the case, then the authors could simply write something like this " The problem was solved by tilting the generator cell in such a way that the pump beam will be reflected at an angle, thereby separating the pump and the Stokes beam".
Use “beam profile” instead of “spot morphology”.
The experimental setup part needs to be re-written.
Lines 175-179: Break the sentence to improve clarity. Have the authors showed the beam combination in Fig. 4? If yes, then they need to specify the components while describing in the text.
Lines 180-183: What were the focal length of the coupling lenses?
Line 187: “Fig. 5(a)” instead of Fig. 5.
Fig.5(b) and (c): Have the authors measured the values of M2 before and after SBS?
Line 204: The extraction efficiency is lower because of the mode mismatch between the pump and the Stokes?
Line 209: A beam quality parameter must be shown. What are the M2 values for these beam profiles shown in the figure?
Line 216: Can the authors provide a Stokes waveform or Stokes energy versus pump plot?
Line 224: How was coherent fringe visibility measured?
Did the experimental results agree with the simulated ones?
A major problem is in this paper is too long sentences. A lot of information is put into one sentence and it’s difficult to follow through. Break the sentences so that there is clarity and helps in understanding the paper. Also, the authors have written pump with capital P in the middle of sentences which is not required. There are places where phrases have been repeated (for example: lines 76-77). So, it would be better if they can re-write the paper with shorter sentences and include more details.
Author Response
Dear Reviewer,
Thank you very much for your letter and advice on our manuscript. It is important to improve the manuscript. Accordingly, we have revised the manuscript. All amendments are underlined in the revised manuscript. In addition, point-by-point responses to the comments are listed below this letter.
We hope that the revised version of the manuscript is now acceptable for publication in your journal. I’m looking forward to hearing from you soon.
Sincerely,
Yu Yu
We would like to express our sincere thanks to the reviewers for the constructive and positive comments.
Replies to the reviewers:
Comment 1: " A major problem is in this paper is too long sentences. A lot of information is put into one sentence and it’s difficult to follow through. Break the sentences so that there is clarity and helps in understanding the paper. Also, the authors have written pump with capital P in the middle of sentences which is not required. There are places where phrases have been repeated (for example: lines 76-77). So, it would be better if they can re-write the paper with shorter sentences and include more details. "
Reply: Thanks for your suggestions. We have rewritten into a few short sentences in the revised manuscript. In detail, “The beam combination system consists of a mirror, a half waveplate and a polarizer. The half-wave plate is used to adjust the polarization state of the two beams with the function of injecting it into the p polarized light together with the unconverted s polarized state. In the meanwhile, it can combine the beam output through the polarizer, and the two beams of light are injected into the position sum in the polarizer.” We have revised the “Pump” into “ pump” in the whole revised manuscript. And places where phrases have been repeated (for example: lines 76-77) have been rewritten in the revised manuscript.
Comment 2: " Lines 8,10: Stimulated Brillouin scattering instead of Brillouin scattering. MOPA is master oscillator power amplifier. Please correct it in the text. Also, I have noticed that the authors have repeated stimulated Brillouin scattering throughout the text. Once you have mentioned the abbreviated form (such as SBS or MOPA), you can simply continue writing the abbreviated form instead of the full form in the subsequent text. "
Reply: Thanks for your suggestions. We have revised the “Brillouin scattering” into “Stimulated Brillouin scattering” in the whole manuscript. And the “SBS” with abbreviated form instead of full form in the subsequent text. And the MOPA has been rewritten into a few short sentences in the revised manuscript. In detail, “The beam combination system consists of a mirror, a half waveplate and a polarizer. The half-wave plate is used to adjust the polarization state of the two beams with the function of injecting it into the p polarized light together with the unconverted s polarized state. In the meanwhile, it can combine the beam output through the polarizer, and the two beams of light are injected into the position sum in the polarizer.” We have revised the “Pump” into “ pump” in the whole revised manuscript. And places where phrases have been repeated (for example: lines 76-77) have been rewritten in the revised manuscript.
Comment 3: " In the introduction, the sentences are too long. Please break the sentences for more clarity. "
Reply: Thanks for your suggestions. We have broken the sentences for more clarity rewritten into a few short sentences in the revised manuscript.
Comment 4: " Line 37: What is thermo-optical wedge? "
Reply: Thanks for your suggestions. Thermo-optical wedge effect caused by optical irradiation on mirrors, generating more heat, especially in the polarized mirror. High power laser irradiation accumulates a large amount of heat in the crystal, leading to uneven heating of the crystal and the generation of deflection effect during laser amplification.
Comment 5: " Line 40: Please provide references. "
Reply: Thanks for your suggestions. A reference [6-8] for coherent synthesis has been added in the revised manuscript.
Comment 6: “Line 49: what is co-frequency resonance?”
Reply: Thanks for your suggestions. Co-frequency resonance is that during the coherent beam combining process of two or more lasers, it is necessary to ensure that each laser frequency is consistent and phase locked.
Comment 7: “Line 65: Please write “compared to” instead of “compared with”.”
Reply: Thanks for your suggestions. We have revised the “compared with” into “compared to” in the revised manuscript.
Comment 8: “Lines 68-71: The advantages for passive phase-locking needs more clarity. It would also be better if shorter sentences are used to describe.”
Reply: Thanks for your suggestions. Compared to the active phase-lock technology, more concisely, the passive method does not need a complex circuit phase control system including phase detection de-vice, strict and complex algorithms. The long sentence has been shortened into two sentences.
Comment 9: “Line 73: A reference for coherent synthesis would be good.”
Reply: Thanks for your suggestions. A reference [] for coherent synthesis has been added in the revised manuscript.
Comment 10: “Lines 77-81: Sentence is too long and needs clarity. Also, the comparison of both the techniques is repeated and not required.”
Reply: Thanks for your suggestions. To avoid repetition in the introduction, we delete the comparison in lines 77-81.
Comment 11: “Line 88: The simulation part needs more explanation. What equations/parameters were used to simulate? How does the generation cell look like?”
Reply: Thanks for your suggestions. We have added more explanation for simulation part. According to the coupled wave equation [Damzen M, Vlad V, Mocofanescu A, and Babin V. Stimulated Brillouin scattering: fundamentals and applications[M]. CRC press, 2003], the electric field components of pump and stokes light can be expressed as the following equations.
(1)
(2)
(3)
where, n is the refractive index of SBS medium, EL and ES represent amplitudes of pump and stokes. And ωL and ωS are corresponding angular frequency angular frequencies. ρ, qB, c, γ, ρ0, ΓB, and α are the density amplitude in SBS medium, the wave vector of the acoustic field, the speed of light, Electrostrictive constant, the density under medium equilibrium state.
(4)
. (5)
The numerical calculation formula can be obtained through the coupled wave equation. In order to achieve better Stokes light waveform output, the generation cell was simulated, and the simulation results are shown in Figure 1.
Comment 12: “Fig. 2: How is the Stokes light reflectance calculated? What is the reason for the shape of the curve?”
Reply: Thanks for your suggestions. The ratio of Stokes light to the pump light injected into the medium pool is the reflectivity of Stokes light. The pump light is focused and injected into the medium pool, and under the action of the grating generated by stimulated Brillouin scattering, a backward Stokes light is generated. The backward Stokes light interacts with the pump light in the medium pool, and the front of the Stokes light quickly absorbs the pump light energy, causing the front of the Stokes light to rapidly increase, leading to corresponding compression of the pulse width. As the energy of the pump light gradually increases, the ability of Stokes light to absorb the pump light gradually increases.
Comment 13: “Fig.3: A diagram to explain the simulation would be helpful.”
Reply: Thanks for your suggestions. A glass tube with a certain tilt angle on the end face is used as the generation pool for Stokes light. Through this change, the pump light reflected by the window surface will be reflected onto the optical platform, and will not be reflected back into the original optical path by the window mirror. The pump light transmitted from the window mirror will generate Stokes light under the grating generated by stimulated Brillouin scattering. Stokes light will be reflected and output along the original path of the pump light path.
Comment 14: “Lines 116-122: Too long a sentence. Please break it to multiple sentences.”
Reply: Thanks for your suggestions. We have broken the long sentence into several short sentences.
Comment 15: “Fig. 4: Mark the beam direction. Please mention what each abbreviated component depicts (such as FR: Faraday rotator…).”
Reply: Thanks for your suggestions. We have added the beam direction in Fig. 4. And added abbreviated components depict as follows, FR : faraday rotator; AMP : amplifier; QP : quarter wave plate; and P : polarizer.
Comment 16: “It is difficult to follow through the experimental setup text since the authors have not specified which mirrors, generator cells they are talking about. Please specify the components in the description so that the reader can understand. For example, in line 48: “Therefore, the seed light path is adjusted as follows, the seed passes through the beam splitter mirror (M1)”. Also, the setup lacks details.”
Reply: Thanks for your suggestions. We specify the components of experimental setup in the description as follows. In the experiment of generating Stokes beam, FC-770 was used as the generation medium [], and the structure of generating Stokes beam was a single SBS cell structure, and the window mirror of the traditional media cell was adopted. Although, this structure can ensure that most of the laser energy passes through the window mirror, and will not produce large refraction after passing through the window mirror, due to the limitation of the coating process, the Pump light energy can not completely pass through the medium cell, and some of the light will still be partially reflected by the window mirror. The surface is reflected, and although the reflected Pump light energy is low, it still affects the returned Stokes beam, resulting in the inability to accurately measure Stokes. The energy and beam profile of the light, so the experiment adopts a new type of media cell structure, that is, the glass tube with a certain inclination angle on the end face is used as the Stokes light production pool, through this change, the Pump light reflected by the window surface will be reflected to the optical platform, and will not be returned to the incident light path by the window mirror. The transmitted Pump light will produce Stokes light through the action of SBS, and Stokes light will be reflected and output along the incident light path, and in this way, it will effectively work. In addition to the effect of the returned Pump light on the returned Stokes light.
Therefore, the seed light path is adjusted as follows, the seed passes through the beam splitter mirror. Particularly, the first beam of light passes through the isolation system, and after passing through a quarter wave plate, the P polarized light is converted into circularly polarized light, and is focused into the SBS cell through a lens, and SBS occurs in the cell to produce a circularly polarized Stokes light with high beam quality. After passing through a half-waveplate again, the Stokes light is converted into P-polarized light, and it is again divided into two beams by a beam splitter (T/R=5:5). After passing through a quarter wave plate, P polarized light is converted into circularly polarized light, injected into two SBS media cells, absorbs the amplified pump energy, and passes through a quarter wave plate again to convert circularly polarized light into S again. The polarized light is output from the entire optical path system, and the output light is adjusted through the optical path and injected into the same polarizer at the same time for the final polarization beam output.
The seed light output by the seed source, the second beam of light after beam splitting by the beam splitter, and the beam is divided into two beams by a beam splitter (T/R=1/1), and the two beams of light are amplified by different amplifiers, and the amplified pump After passing through a polarizer and a quarter waveplate, the light changes from P-polarized light to S-polarized light, which is injected into the SBS medium pool as Stokes. The pump light of the light, the pump light is amplified by different amplifiers, and the amplified pump light is converted into circularly polarized light by a quarter wave plate respectively in the media cell to achieve collinear amplification, and the pump light that cannot be extracted passes through the media cell. After being converted into S-polarized light through a quarter wave plate again, it is reflected out of the optical path by the polarizer and enters the light absorption system. The amplified Stokes light is converted into S-polarized light by a quarter waveplate, reflected through the polarizer, and output through the final beam clamping system.
Comment 17: “The authors need to describe the length of the generator cells. FC 770 is used as the SBS medium. A reference is needed for the material. Please specify the Brillouin shift and expected Stokes wavelength. What are AMP1 and AMP2? How does SBS-PCM works? Please provide references for it too.”
Reply: Thanks for your suggestions. In the experimental setup, the focal length of the lens and the length of SBS cell are 80 cm and 60 cm, respectively. In the experiment of generating Stokes beam, FC-770 was used as the generation medium [5], with the parameters including SBS medium refractive index of 1.27, phonon lifetime of 0.57 ns, gain coefficient of 3.5 cm/GW and the Brillouin frequency shift of 1081 MHz, respectively. Merely with the Brillouin frequency shift of 1.081 GHz, the expected Stokes wavelength is still round at 1064 nm. AMP1 and AMP2 represent the amplifies with the function of realizing the amplification of two Stokes beams. The pump light transmitted from the window mirror will generate Stokes light under the grating generated by stimulated Brillouin scattering. Stokes light with the conjugate phase will be reflected and output along the original path of the pump light path. Functionally, the spot morphology of the injected pump light was improved to some extent by the action of SBS-PCM, and the energy distribution of the laser spot output became more uniform. We provide reference [5] for verifying it.
Comment 18: “Lines 139-144: Was this method used to separate the pump and the Stokes beam? If that is the case, then the authors could simply write something like this " The problem was solved by tilting the generator cell in such a way that the pump beam will be reflected at an angle, thereby separating the pump and the Stokes beam". Use “beam profile” instead of “spot morphology”. The experimental setup part needs to be re-written.”
Reply: Thanks for your suggestions. Due to the limitation of the coating process, the pump light energy can not completely pass through the medium cell, and some of the light will still be partially reflected by the window mirror. To avoid the return light of the pump from entering the oscillator, resulting in the inability to accurately measure the energy and beam profile of the Stokes light. Therefore, a new type of media cell structure was adopted, that is, the glass tube with a certain inclination angle on the end face is used as the Stokes light production pool. We have revised the “beam profile” into “spot morphology” in the whole manuscript. And the experimental setup part has been re-written.
Comment 19: “Lines 175-179: Break the sentence to improve clarity. Have the authors showed the beam combination in Fig. 4? If yes, then they need to specify the components while describing in the text.”
Reply: Thanks for your suggestions. Due to the limitation of the coating process, the pump light energy cannot completely pass through the medium cell, and some of the light will still be partially reflected by the window mirror. To avoid the return light of the pump from entering the oscillator, resulting in the inability to accurately measure the energy and beam profile of the Stokes light. Therefore, a new type of media cell structure was adopted, that is, the glass tube with a certain inclination angle on the end face is used as the Stokes light production pool. We have revised the “beam profile” into “spot morphology” in the whole manuscript. And the experimental setup part has been re-written as follows. The beam combination system in the experiment consists of a mirror, a half wave-plate and a polarizer. In detail, the half-wave plate is used to adjust the polarization state of each beam. Functionally, it can convert the S-polarized light into P-polarized light. After injecting into the polarizer together with the unconverted S polarized light, the combined beam output with the same phase, and the two Stokes beams after amplification are injected in the polarizer. The optical path difference can be adjusted by a set of coupling lenses, which are fixed on a set of sliding tables. In order to manipulate the optical path difference, the coupling lens group is ad-justed by moving the position of the frame on the slide table. Therefore, as to change the optical path difference and adjust the two beams of light, the optimal beam has been fixed through the position of coupling lenses.
Comment 20: “Lines 180-183: What were the focal length of the coupling lenses?”
Reply: Thanks for your suggestions. It does not require the use of a coupling lens, but rather adjusting the optical path difference by fixing a set of highly reflective flat mirrors on a sliding table, and adjusts the reflecting mirror group by moving the sliding table to change the optical path difference and adjust the two beams of light to the optimal beam closure position.
Comment 21: “Line 187: “Line 187: “Fig. 5(a)” instead of Fig. 5.”
Reply: Thanks for your suggestions. Figure 5 has been revised into Figure 5(a). The annotation in Fig. 5. (a) has been revised into “The Stokes light energy and energy reflectivity after the generation of SBS-PCM phase conjugate mirror.”
Comment 22: “Fig.5(b) and (c): Have the authors measured the values of M2 before and after SBS?”
Reply: Thanks for your suggestions. Unfortunately, we haven’t measured the value of M2 before and after SBS. And the entire experimental setup has been dismantled.
Comment 23: “Line 204: The extraction efficiency is lower because of the mode mismatch between the pump and the Stokes?”
Reply: Thanks for your suggestions. Due to the influence of thermal effect, the laser center of the amplified Pump becomes unstable. As a result, Stokes light cannot extract more Pump energy, and the extraction efficiency of Stokes light energy is getting lower and lower.
Comment 24: “Line 209: A beam quality parameter must be shown. What are the M2 values for these beam profiles shown in the figure?”
Reply: Thanks for your suggestions. Unfortunately, we haven’t measured the value of M2 before and after SBS. And the entire experimental setup has been dismantled.
Comment 25: “Line 216: Can the authors provide a Stokes waveform or Stokes energy versus pump plot?”
Reply: Two Stokes signal beams, before participating in beam combining, received the influence of Pump light, which resulted in poor quality of the amplified output spot beam. However, after coherent beam combining, strong interference fringes were still generated. This indicates that it is feasible to divide one Stokes beam into two beams and use the principle of backward seed injection passive phase-locked technology. Low energy Stokes signal light with the same wavelength as the pump light was generated in the generation pool, In the amplification medium pool, the backward injected Stokes light interacts with the pump light to achieve passive phase locking.
Comment 26: “Line 224: How was coherent fringe visibility measured? Did the experimental results agree with the simulated ones?”
Reply: Thanks for your suggestions. By adjusting the position of the two mirrors on the slide rail in the beam combining system, when the optical path difference between the two sub beams is less than 2cm, dense coherent fringes can be observed in the light spot morphology map by observing the light spot reflected by the wedge mirror. However, due to the strong thermal effect of the pump light at high energy, it causes significant distortion and fluctuations in the pump light. When achieving collinear amplification, Causing significant distortion on the periphery of Stokes light. By measuring and calculating the coherent fringes after combining the two beams, the contrast of the fringes formed after combining the two beams is 87.5%. Therefore, experimental results agree with the simulated ones.
Comment 27: “A major problem is in this paper is too long sentences. A lot of information is put into one sentence and it’s difficult to follow through. Break the sentences so that there is clarity and helps in understanding the paper. Also, the authors have written pump with capital P in the middle of sentences which is not required. There are places where phrases have been repeated (for example: lines 76-77). So, it would be better if they can re-write the paper with shorter sentences and include more details.”
Reply: Thanks for your suggestions. We have rewritten into a few short sentences in the revised manuscript.

Reviewer 5 Report
The authors demonstrated a new technique presented for coherent beam combination based on SBS passive phase-lock. With the advantage of combined scheme, the energy extraction efficiency of SBS coherent combination reached at 91.8% with coherent fringe visibility of 83%. This is a very meaningful result and has good reference value for researchers in related fields. However, it can be published after minor revisions.
1. Though the introduction does include some brief background, it does not provide a sense of the current state-of-the-art or the main problems that need to be addressed. The authors should make some necessary supplements.
2. In order to reduce the influence of thermal effects on beam quality, beam split-amplification has been adopted with the same phase locking by the back injection of Stokes pulse. The authors should explain passive phase-lock effect more clearly.
3. In the whole manuscript, more detail, “Pump light” with upper text and “pump light” with lower text should be same.
Minor editing of English language required.
Author Response
Dear Reviewer,
Thank you very much for your letter and advice on our manuscript. It is important to improve the manuscript. Accordingly, we have revised the manuscript. All amendments are underlined in the revised manuscript. In addition, point-by-point responses to the comments are listed below this letter.
We hope that the revised version of the manuscript is now acceptable for publication in your journal. I’m looking forward to hearing from you soon.
Sincerely,
Yu Yu
The authors demonstrated a new technique presented for coherent beam combination based on SBS passive phase-lock. With the advantage of combined scheme, the energy extraction efficiency of SBS coherent combination reached at 91.8% with coherent fringe visibility of 83%. This is a very meaningful result and has good reference value for researchers in related fields. However, it can be published after minor revisions.
Comment 1: “Though the introduction does include some brief background, it does not provide a sense of the current state-of-the-art or the main problems that need to be addressed. The authors should make some necessary supplements.”
Response: Thanks for your suggestions. In the introduction part, current state-of-art researches have been added as follows. Based on corner cube, Cheng Yong. et al. [17] utilized mutual injection phase-locked technology to realize 6-channel laser coherent synthesis. It exhibited that the ener-gy of the output light is 15.3 J with the pulse width of 500 μs, the divergence angle is 1.7 mrad, and the synthesis efficiency is as high as 95.6%. Utilized self-phase locking technology, Hongjin Kong et al. [18] adopted wavefront segmentation of pump pulses and combining with SBS-PCM. By using piezoelectric ceramic (PZT) feedback to adjust the distance between the concave mirror and the SBS cell, the coherent synthesis of four beams was achieved. However, it still requires detection and modulation of the phase of the laser array, and real-time adjustment of the PZT feedback mirror at the end of the SBS-PCM in each laser path based on the test and calculation results to achieve coherent laser output, which greatly increases the complexity of the locking system.
Comment 2: “In order to reduce the influence of thermal effects on beam quality, beam split-amplification has been adopted with the same phase locking by the back injection of Stokes pulse. The authors should explain passive phase-lock effect more clearly.”
Response: Thanks for your suggestions. Compared to the active phase-lock technology, more concisely, the passive method does not need a complex circuit phase control system including phase detection de-vice, strict and complex algorithms. The overall structure is complex and needs to occupy more space, and passive phase-locking technology is a method of self-phase adjustment. Through the structure of some optical structures or optical characteris-tics, and finally it realizes the phase-locking method of two beams or multiple beams. Therefore, we demonstrate a passive phase locking coherent combination method of solid lasers through stimulated Brillouin scattering effect. Meanwhile, it paves the way for excellent performances of lasers including high pulse energy, and repetition frequency.
Comment 3: In the whole manuscript, more detail, “Pump light” with upper text and “pump light” with lower text should be same.
Response: Thanks for your suggestions. We have revised the “Pump” into “ pump” in the whole revised manuscript.

Round 2
Reviewer 2 Report
The authors have significantly improved the paper, so, I can recommend it for publication.
Reviewer 4 Report
The authors have adequately replied to my queries.
The paper looks good.
It is much better than last time.